# Antitumor Effects of *N*-Butylidenephthalide Encapsulated in Lipopolyplexs in Colorectal Cancer Cells

**DOI:** 10.3390/molecules25102394

**Published:** 2020-05-21

**Authors:** Kai-Fu Chang, Jinghua Tsai Chang, Xiao-Fan Huang, Yu-Ling Lin, Kuang-Wen Liao, Chien-Wei Huang, Nu-Man Tsai

**Affiliations:** 1Institute of Medicine, Chung Shan Medical University, Taichung 40201, Taiwan; kfchang1015@gmail.com (K.-F.C.); jinghuat@csmu.edu.tw (J.T.C.); s9870509@gmail.com (X.-F.H.); 2Department of Medical Laboratory and Biotechnology, Chung Shan Medical University, Taichung 40201, Taiwan; 3Agricultural Biotechnology Research Center, Academia Sinica, Taipei 11529, Taiwan; lyring@gate.sinica.edu.tw; 4Department of Biological Science and Technology, National Chiao Tung University, Hsinchu 30068, Taiwan; liaonms@pchome.com.tw; 5Institute of Molecular Medicine and Bioengineering, National Chiao Tung University, Hsinchu 30068, Taiwan; 6Division of Gastroenterology, Department of Internal Medicine, Kaohsiung Armed Forces General Hospital, Kaohsiung 80284, Taiwan; 7Clinical Laboratory, Chung Shan Medical University Hospital, Taichung 40201, Taiwan

**Keywords:** *N*-butylidenephthalide, LPPC nanoparticle, colorectal cancer, antitumor, cytotoxicity

## Abstract

Colorectal cancer (CRC) is the third most common type of cancer and the second most common cause of cancer-related death in the world. *N*-Butylidenephthalide (BP), a natural compound, inhibits several cancers, such as hepatoma, brain tumor and colon cancer. However, due to the unstable structure, the activity of BP is quickly lost after dissolution in an aqueous solution. A polycationic liposomal polyethylenimine and polyethylene glycol complex (LPPC), a new drug carrier, encapsulates both hydrophobic and hydrophilic compounds, maintains the activity of the compound, and increases uptake of cancer cells. The purpose of this study is to investigate the antitumor effects and protection of BP encapsulated in LPPC in CRC cells. The LPPC encapsulation protected BP activity, increased the cytotoxicity of BP and enhanced cell uptake through clathrin-mediated endocytosis. Moreover, the BP/LPPC-regulated the expression of the p21 protein and cell cycle-related proteins (CDK4, Cyclin B1 and Cyclin D1), resulting in an increase in the population of cells in the G_0_/G_1_ and subG_1_ phases. BP/LPPC induced cell apoptosis by activating the extrinsic (Fas, Fas-L and Caspase-8) and intrinsic (Bax and Caspase-9) apoptosis pathways. Additionally, BP/LPPC combined with 5-FU synergistically inhibited the growth of HT-29 cells. In conclusion, LPPC enhanced the antitumor activity and cellular uptake of BP, and the BP/LPPC complex induced cell cycle arrest and apoptosis, thereby causing death. These findings suggest the putative use of BP/LPPC as an adjuvant cytotoxic agent for colorectal cancer.

## 1. Introduction

Angelica sinensis root (Danggui) is one of the most commonly used herbs in traditional Chinese medicine (TCM), and is consumed a dietary supplement in Europe and North America [1]. *N*-Butylidenephthalide (BP) is the active component of Angelica sinensis; it has a molecular weight of 188.23 g/mol. The biological functions of BP include the induction of angiogenesis [2], inhibition of smooth muscle cells in mice [3] and inhibition of the NF-κB pathway to decrease interleukin-6 and TNF-α secretion from dendritic cells [4]. In terms of the antitumor effects, BP inhibits the growth of different types of cancer, such as colon cancer, brain tumor, hepatoma, and lung cancer [5,6,7,8]. Additionally, BP induces p53-dependent and -independent apoptosis, regulates the levels of cyclin kinase inhibitors such as p21 and p27, and inhibits the ERK1/2 pathway to prevent tumor growth [6,8]. The structure of BP is hydrated or oxidized after dissolution in solution, which may affect the antitumor activity of BP [9,10,11,12]. Moreover, BP is quickly metabolized by liver cells, and 80% of BP metabolites are excreted in the urine within 24 h, resulting in a short half-life (~12 h) in rats [10]. The activity of BP must be preserved prior to its use in cancer therapy.

According to a recent study, colorectal cancer (CRC) is the fourth most common cancer and the third leading cause of death in the United States [13,14]. An estimated 1.2 million new cases of CRC are diagnosed worldwide, and up to 30% of patients with CRC have poor prognoses due to local recurrence or distant metastasis [15,16]. After surgery, chemotherapy is required to treat patients with CRC; however, the standard cytotoxic chemotherapeutic drugs, including 5-fluorouracil, oxorubicin and mitomycin, nonspecifically enter normal organs and induce side effects [17,18]. Therefore, a new drug that is highly selective for colorectal cancer cells is needed.

Studies confirmed that liposomes as drug carriers reduce side effects by decreasing drug penetration into normal organs, slowly releasing drugs, maintaining drug stability and increasing the drug uptake rate of cancer cells [19,20,21,22,23]. The liposome also increases apoptosis in cancer cells in vitro and in vivo [24] and enhances antitumor effects on pancreatic cancer, hepatocellular carcinoma and osteosarcoma [25,26,27]. In addition, the cationic liposome carries negatively charged molecules, such as DNA, RNA and peptides, into cells [28,29], and triggers cell endocytosis to increase drug uptake [30,31,32].

Polycationic Liposome Containing PEI and Polyethylene Glycol Complex (LPPC), a novel liposome, was developed by Dr. Kuang-Wen Liao and colleagues at National Chiao Tung University. LPPC is a lipid bilayer composed of DOPC and DLPC and noncovalently modified with PEG and PEI [29]. PEG prolongs the circulation and reduces inflammatory reactions in vivo [33]. LPPC exhibits a cationic zeta potential due to the presence of PEI with a strong positive charge, which increases cellular uptake [32]. Moreover, LPPC was used to encapsulate curcumin and form a curcumin/LPPC complex, and it inhibited drug-resistant cell growth and suppressed tumor growth in vivo [34]. Additionally, the LPPC delivery system shows improved drug transport that may be beneficial as an innovative tool for cancer therapy [35,36]. As shown in our previous studies, LPPC protects BP from oxidation and enhances antitumor effects on glioblastoma and melanoma [37,38]. In this study, we aimed to investigate the protective effect of LPPC on BP activity and the antitumor effects of BP encapsulated in LPPC in colorectal cancer.

## 2. Results and Discussion

### 2.1. Effect of BP/LPPC on Cell Viability in Colorectal Cancer Cells

BP/LPPC inhibits the growth of GBM and melanoma cells [38,39], but its effects in CRC cells are unknown. First, the viability of BP/LPPC- or BP-treated cells was detected using a MTT assay to determine the inhibitory effect in CRC cells. Both BP/LPPC and BP inhibited the growth of cancer cells in a dose-dependent manner at 24 and 48 h (Figure 1a). The IC_50_ values of normal cells in the BP/LPPC group (24.15 ± 0.40–26.74 ± 3.82 μg/mL, 24 h) were significantly higher than those of CRC cells (9.61 ± 2.97–11.01 ± 3.96 μg/mL, 24 h), indicating that BP/LPPC exhibited higher selectivity toward CRC cells (Table 1). Additionally, the CRC cells were more sensitive to BP/LPPC (9.61 ± 2.97–11.01 ± 3.96 μg/mL, 24 h) than BP (47.87 ± 2.3–73.91 ± 2.98 μg/mL, 24 h) or BP/liposomes (69.61 ± 1.74 –139.33 ± 2.32 μg/mL, 24 h). The ratio of the IC_50_ values for BP/LPPC to BP was 4.3–7.7, and the ratio of BP/LPPC to BP/liposome was 6.3–14.5. Based on these results, BP/LPPC displayed greater cytotoxicity toward CRC cells than BP and BP/Liposome, but lower cytotoxicity toward normal cells than CRC cells. Thus, the LPPC encapsulation increased the cytotoxicity of BP and selectivity toward CRC cells. Moreover, BP/LPPC exerted a more toxic effect than BP/Liposome, suggesting that the encapsulation of BP with LPPC was more suitable for this drug. Because of its unstable structure, BP may lose its activity during the preparation of the BP/Liposome using traditional liposome encapsulation methods. LPPC encapsulation protected the unstable structure of BP, maintained its bioactivity in the environment and increased its cytotoxicity.

### 2.2. LPPC Encapsulation Stabilized BP Activity for Cytotoxicity of Colorectal Cancer Cells

Next, the drug designs included LPPC encapsulation (BP/LPPC group), no LPPC encapsulation (BP group) and BP + empty LPPC (BP + LPPC group) to determine whether the activity of BP was protected after encapsulation in LPPC. In HT-29 and CT26 cells, BP/LPPC showed greater cytotoxicity (IC_50_ = 11.06 ± 0.37–27.60 ± 1.10 μg/mL, 24 h) than BP (IC_50_ = 145.32 ± 0.35–213.41 ± 2.04 μg/mL, 24 h) and BP + LPPC (IC_50_ = 121.6 ± 6.64–176.81 ± 4.56 μg/mL, 24 h) after storage at 4 °C in H_2_O (Figure 1b,c). In addition, BP/LPPC (IC_50_ = 14.57 ± 0.15–38.38 ± 5.91 μg/mL, 24 h) also displayed greater cytotoxicity than the BP group (IC_50_ = 138.03 ± 2.88–173.25 ± 0.52 μg/mL, 24 h) and BP + LPPC (IC_50_ = 155.02 ± 2.96–188.14 ± 0.3 μg/mL, 24 h) after storage at 37 °C in PBS containing 10% FBS (Figure 1d,e). The IC_50_ value was rapidly increased in the BP group and BP + LPPC group after an incubation at 4 °C or 37 °C for 4–24 h, but was not obviously altered in the BP/LPPC groups. The structure of BP was reported to be easily hydrated or oxidized, and thus, the biological functions of BP may be altered or the activity lost after dissolution in an aqueous solution. However, the BP activity was maintained or increased in the BP/LPPC group, suggesting that LPPC encapsulation stabilized the BP structure and improved its antitumor activity.

### 2.3. LPPC Encapsulation Increased Cell Uptake of BP through Induction of Clathrin-Mediated Endocytosis

Previous studies of liposomes revealed that liposomes decrease drug penetration into normal organs, maintain drug stability and increase cellular uptake [19,20,21,22,23]. Next, we quantitatively and qualitatively investigated whether LPPC encapsulation promotes the uptake of BP in CRC cells. After drug treatment, the BP fluorescence was observed in cells in the BP/LPPC group at 15 min and in the BP group at 60 min (Figure 2a). The BP values of cell uptake in the BP/LPPC group (12.78 ± 0.22–20.37 ± 1.21 μg/2.5 × 10^5^ cells) were greater than in the BP group (1.42 ± 0.01–7.97 ± 2.17 μg/2.5 × 10^5^ cells) from 15 to 60 min after treatment (Figure 2b), indicating that LPPC encapsulation increased the rate of BP uptake in CRC cells.

Liposomes with a positive charge trigger endocytosis to increase cellular uptake [40,41]. In our previous study, the average zeta potential of BP/LPPC was ~38 mV [37], which may induce cell endocytosis. Cells were pretreated with the endocytosis inhibitors AHH (micropinocytosis), FIII (caveolae-mediated endocytosis) or CPZ (clathrin-mediated endocytosis) prior to the BP/LPPC treatment to determine which endocytosis pathway was involved in BP/LPPC uptake. The cells were collected, and the BP levels were measured; all inhibitors reduced the cellular uptake of BP compared with the control group (12.78 ± 0.22–19.71 ± 0.24 μg/2.5 × 10^5^ cells) from 15 to 90 min, particularly in the CPZ groups (1.86 ± 0.03–3.30 ± 0.02 μg/2.5 × 10^5^ cells; Figure 2c). LPPC encapsulation triggered cellular endocytosis to increase the uptake of BP into CRC cells through the clathrin-mediated endocytosis pathway. However, the phenomenon was not observed in normal cells (data not shown), potentially due to the differences in the characteristics of normal and cancer cells. Furthermore, this property of LPPC may be useful for distinguishing normal and cancer cells to reduce drug-related side effects during therapy.

### 2.4. BP/LPPC Induce Cell Cycle Arrest and Cell Apoptosis in Colorectal Cancer Cells

Cells were treated with BP/LPPC (30, 60 or 90 μg/mL) or BP (100, 140 or 180 μg/mL) and the cell cycle was analyzed by monitoring the FL2 intensity using a FACScan instrument to examine the antitumor mechanism. Treatment with 60 μg/mL BP/LPPC (1–12 h) and 140 μg/mL BP (6–12 h) induced cell cycle arrest at the G_0_/G_1_ phase (62.57 ± 0.42% in BP/LPPC at 1 h; 54.22 ± 0.53% in BP at 6 h, Figure 3a). In addition, BP/LPPC (30, 60 and 90 μg/mL) and BP (180 μg/mL) induced G_0_/G_1_ phase arrest after treatment for 24 h (Figure 3b). Therefore, both BP/LPPC and BP induced cell cycle arrest to inhibit the growth of CRC cells. By analyzing the cell cycle, the percentage of cells in the subG_1_ phase was increased after BP/LPPC and BP treatment in time- and dose-dependent manners (Figure 3c,d). After treatment with BP/LPPC (60 μg/mL) for 6 h or BP (140 μg/mL) for 24 h, cell apoptosis was analyzed using the TUNEL assay and observed with a fluorescence microscope at ×400 magnification. Treated cells were TUNEL-positive (81.53 ± 4.33% in BP/LPPC; 75.10 ± 3.15% in BP) and exhibited an apoptotic morphology, including chromatin condensation, DNA fragmentation and apoptotic bodies (Figure 3e,f).

### 2.5. BP/LPPC Regulated Cell Cycle and Apoptosis Associated Protein Expression

Western blotting results revealed decreased levels of cell cycle-related proteins (p-Rb and Rb) and increased levels of a cell cycle regulator (p21). In addition, the levels of proteins involved in regulating the cell cycle, including CDK4/Cyclin D1 (G_0_/G_1_ phase) and CKD2/Cyclin B (G_2_/M phase), were decreased after BP/LPPC and BP treatment (Figure 4a,b). The levels of apoptosis-associated proteins were analyzed using western blotting to examine the cell death pathway. The levels of proteins involved in the extrinsic (Fas, FasL and Caspase-8) and intrinsic apoptosis pathways (Bax and Caspase-9) were increased and activated after the BP/LPPC or BP treatment (Figure 4a,b). Finally, Caspase-3 was activated and initiated the caspase signal transduction cascade. At 6 h, BP/LPPC (60 μg/mL) induced greater activation of Caspase-8 and Caspase-3 than BP (140 μg/mL; Figure 4c). Additionally, the activation of Caspase-3 induced by BP/LPPC or BP was blocked when cells were pretreated with a Caspase-3 inhibitor (Figure 4d). BP/LPPC and BP induced both the extrinsic and intrinsic apoptosis pathways, triggered the caspase cascade and induced cell death. At the same time, BP/LPPC (60 μg/mL) exerted a better antitumor effect and efficacy than BP (140 μg/mL) on inhibiting cell proliferation, inducing cell cycle arrest and cell apoptosis, suggesting that BP/LPPC should be considered a potential therapeutic agent for treating CRC.

### 2.6. Combination of BP/LPPC and 5-FU Synergistically Inhibits the Growth of Colorectal Cancer Cells

The first-line CRC therapy is 5-fluorouracil (5-FU), which has confirmed antitumor potential because it regulates thymidylate synthase activity and decreases the synthesis of DNA and RNA [42]. The survival benefits of 5-FU have been established; however, the clinical applications of this drug have been limited because of the development of drug resistance and adverse side effects after treatment with higher doses. Recent progress has been achieved in employing novel combination therapeutic strategies to improve the treatment efficacy and moderate adverse effects [43]. Therefore, an experiment was designed in which cells were treated with BP/LPPC combined with 1 μg/mL 5-FU or 5-FU combined with 10 μg/mL BP/LPPC to determine the potential synergistic effects of BP/LPPC and 5-FU. Cell viability was decreased in the group treated with BP/LPPC combined with 5-FU (51.27 ± 0.90 and 47.84 ± 0.66 μg/mL) to a greater extent than the group treated with BP/LPPC alone (71.67 ± 0.78 and 56.18 ± 0.16 μg/mL) at 2.5 and 5 μg/mL doses, respectively (Figure 5a). Additionally, the viability of cells treated with 5-FU combined with BP/LPPC (44.97 ± 1.70 and 42.05 ± 1.37 μg/mL) was reduced compared with cells treated with 5-FU alone (79.43 ± 1.26 and 70.95 ± 7.04 μg/mL) at 0.5 and 1 μg/mL doses, respectively (Figure 5b). The combination index (CI) was 0.50 in cells treated with the combination of BP/LPPC and 5-FU, suggesting that BP/LPPC combined with 5-FU exerted a synergistic effect. In CRC, 5-FU is commonly used in chemotherapy, but simultaneously induces side effects in normal organs. As shown in the present study, the combination of BP/LPPC and 5-FU exerted a synergistic effect, and thus, may reduce the required dose of 5-FU, suggesting that the side effects would be reduced.

CRC is one of the most common malignancies worldwide, with particularly high incidences in developed countries [44]. Surgical ablation, radiotherapy and conventional chemotherapy are the major strategies for treatment of CRC, but none of these is completely effective because of general toxicity and size effects. Therefore, new therapeutic agents with better efficiency and fewer side effects are urgently needed for the treatment of CRC. Recent studies demonstrated that liposomes improve the efficiency of drugs by increasing the rate of drug uptake, maintaining drug stability and slowly releasing drugs [19,20,21,22,23]. In this study, we demonstrated that LPPC encapsulation protected BP activity, enhanced cellular uptake and increased selection to CRC cells, all of which will improve the efficiency of BP treatment. According to reports, cell cycle deregulation is one of the hallmark features of cancer cells. The loss of cell cycle checkpoints before completing DNA repair will activate the apoptosis cascade, thereby causing cell death [45,46]. Therefore, drugs which induce apoptosis or cell cycle arrest would be an excellent source of new anticancer agents. The results demonstrated that BP/LPPC induced cell cycle arrest at the G_0_/G_1_ phase by upregulation of p53 and p21 protein expression and downregulation of CDK4/Cyclin D1 protein expression. Moreover, BP/LPPC increased the percentage of the subG_1_ phase, and activated extrinsic (FasL/Fas/Caspase-8) and intrinsic (Bax/Caspase-9) apoptosis pathways, contributing to cell death. These findings further confirm the therapeutic potential of BP/LPPC in CRC.

## 3. Materials and Methods

### 3.1. Cell Culture and Reagents

Two types of colon cancer cells, HT-29 and CT26, and two normal cell lines, MDCK and SVEC, were used in the present study. Those cells were obtained from the Food Industry Research and Development Institute (Hsinchu, Taiwan). The culture media included Dulbecco’s Modified Eagle’s Medium (HT-29, MDCK and SVEC cells) and RPMI-1640 (CT26) containing 10% heat inactivated fetal bovine serum (Gibco BRL, Gaithersburg, Maryland), HEPES (10 mM; Gibco), pyruvate (1 mM; Gibco) and P/S (100 U/mL penicillin and 100 μg/mL streptomycin; Gibco). All cells were subcultured with 0.25% trypsin/0.53 mM EDTA solution, and maintained in growth medium in a humidified atmosphere with 5% CO_2_ at 37 °C. TP53, BRAF, KRAS and PIK3CA were mutated in HT-29 cells, as detected using the automated extraction of nucleic acids (AccuBioMed Co, Ltd., Taipei, Taiwan) and Femtopath Human Primer Sets (HongJing Biotech, Taipei, Taiwan). *n*-Butylidenephthalide (BP, purity ≥95%, Alfa Aesar, Thermo Fisher Scientific, Waltham, MA, USA) and 5-fluorouracil (purity ≥99%, Sigma, USA) were dissolved in DMSO and stored at 4 or −20 °C until use in each in vitro experiment. BP was encapsulated in LPPC (from the laboratory of Dr. Kuang-Wen Liao, National Chiao Tung University, Taiwan) using a procedure [37]. The scheme and preparation of LPPC were described in a previous study [29,34].

### 3.2. Analysis of the Cytotoxicity

Cell viability was detected using the MTT assay. Cells were incubated in 96-well plates (5 × 10^3^ cells per well) overnight, and treated with BP/LPPC (0–100 μg/mL) or BP (0–400 μg/mL) dissolved in medium for 24 h. After incubation, the medium was removed, replaced with 100 μL of the MTT solution (400 μg/mL, Sigma), incubated for 6–8 h and replaced with 50 μL of DMSO to dissolve the formazan crystals. The absorbance was measured with a microplate reader (Molecular Devices, Spec384) at 550 nm. The untreated cells served as the control for 100% cell viability and were used to calculate the IC_50_ values.

### 3.3. Detection of BP Activity

BP/LPPC, BP or BP + LPPC were dissolved in 200 μL of H_2_O at 4 °C or 10% FBS in PBS at 37 °C (all BP concentrations were 3 mg/mL) and incubated for 0, 4, 8 or 24 h. Cells were seeded in 96-well culture plates and treated with the preincubated BP/LPPC, BP or BP + LPPC. Cell viability was determined using the MTT assay and IC_50_ values were calculated. The activity of BP/LPPC, BP or BP + LPPC was estimated by measuring the cytotoxicity toward cancer cells.

### 3.4. The Cellular Uptake of BP/LPPC

HT-29 cells (5 × 10^5^ cells) were seeded on 15 mm microscope cover glasses (Assistant, Germany) in 3.5 cm^2^ dishes. After an overnight incubation, the medium was removed from the dishes and cells were treated with BP/LPPC (50 μg/mL) or BP (50 μg/mL) for 0, 15, 30, 45 or 60 min. The cells were washed with PBS, fixed with 10% neutral formalin, and observed with an upright fluorescence microscope (ZEISS AXioskop2, Carl Zeiss, Thornwood, NY, USA) at a magnification of ×400.

HT-29 cells were incubated in 24-well culture plates (2.5 × 10^5^ cells/well) containing 500 μL of growth medium overnight. After removing the medium, the cells were treated with BP/LPPC (50 μg/mL) or BP (50 μg/mL) for 0, 15, 30, 45 or 60 min. After cells were collected, the BP in the cells was extracted with phenol-chloroform and its concentrations were detected with a fluorescence spectrophotometer (HITACHI F-4500, Tokyo, Japan) at 350 nm.

### 3.5. Analysis of Endocytosis

HT-29 cells (2.5 × 10^5^ cells) were plated in each well of 24-well culture plates in 500 μL of growth medium and incubated overnight. The medium was replaced with 300 μL of medium containing different endocytosis inhibitors, including amiloride hydrochloride hydrate (AHH, 13.31 μg/mL, Sigma, USA), filipin III (FIII, 1 μg/mL, Sigma, USA) or chlorpromazine hydrochloride (CPZ, 10 μg/mL, Sigma, USA). The medium was removed after incubation for 1 h, and BP/LPPC (50 μg/mL) or BP alone (50 μg/mL) were added to each well and incubated with the cells for 0, 15, 30, 45 or 60 min. Finally, the amount of BP in the cells was calculated using the method described above.

### 3.6. Cell Cycle Analysis

HT-29 cells (2 × 10^6^ cells) were incubated in 10 cm^2^ dishes overnight, and then treated with BP/LPPC (30, 60 and 90 μg/mL) for 0, 1, 3, 6 or 12 h, or BP (100, 140 and 180 μg/mL) for 0, 6, 12, 24 or 48 h. The cell cycle was evaluated by staining the DNA with propidium iodide (PI) and flow cytometry. The cells were harvested, resuspended in 800 μL of PBS and incubated with 100 μL of PI (400 μg/mL, Sigma-Aldrich, St. Louis, MO, USA) and 100 μL of RNase (1 mg/mL, Sigma). After an overnight incubation at 4 °C, the cells were analyzed for FL2 intensity using a FACScan instrument (Becton Dickinson, USA) and Kaluza Flow Cytometry Analysis Software (Software Version 1.2, Beckman Coulter, USA).

### 3.7. TUNEL Assay

Apoptosis was assayed using an In Situ Cell Death Detection Kit, POD (Roche, Mannheim, Germany), according to the manufacturer’s instructions. The treated cells were fixed with 10% formaldehyde at room temperature for 10 min. Then, the cells were dried on silane-coated glass slides (MATSUNAMI, Tokyo, Japan), rehydrated with PBS, and endogenous peroxidases were inactivated with 3% H_2_O_2_. The cells were incubated with a cold permeabilization solution (0.1% Triton X-100 in 0.1% sodium citrate), incubated with the TUNEL reaction mixture for 2 h at 37 °C, and counterstained with PI (10 μg/mL). The morphology of apoptotic cells was observed under a fluorescence microscope (ZEISS AXioskop2, Carl Zeiss, Thornwood, NY, USA) at ×400 magnification.

### 3.8. Western Blotting

Sample were collected, lysed in RIPA buffer, and protein concentrations were measured with a bicinchoninic acid (BCA) protein assay kit (Pierce, Rockford, IL, USA). All samples (20 μg protein/well) of cell lysates were separated using 10–12.5% sodium dodecyl sulfate–polyacrylamide gel electrophoresis (SDS–PAGE; Tanon, Shanghai, China) and transferred to polyvinylidenedifluoride (PVDF) membranes (FluoroTrans, PALL, Dreieich, Germany). After blocking with 5% skin milk for 1 h at 25 °C, the PVDF membranes were incubated with primary antibodies in BSA solution (1% BSA in TBS-T containing 1% Tween-20) overnight. The antibodies included antiRb, antiphospho-Rb, antip21, antiCKD2, antiCKD4, antiCyclin B1, antiCyclin D1, antiFas, antiFasL, antiBax, antiCaspase-3, antiCaspase-8, antiCaspase-9 and antiβ-actin (Santa Cruz, CA, USA). After washes with TBS-T, the membranes were incubated with the respective horseradish peroxidase-conjugated antimouse, antirabbit or antigoat IgG secondary antibodies (Santa Cruz, CA, USA) for 2 h at 25 °C. The membranes were incubated with the enhanced chemiluminescence (ECL) reagent (T-Pro Biotechnology, Taiwan) and protein expression levels were detected using a chemiluminescence/fluorescence imaging analyzer (GE LAS-4000, GE Healthcare Life Sciences, NJ, USA). Protein levels = (sample intensity/sample β-actin intensity)/(control intensity/control β-actin intensity).

### 3.9. Analysis of Caspase-3 Activity

Cells were seeded in 6-well culture plates (5 × 10^5^ cells/well) and incubated overnight. After removing the medium, the cells were pretreated with the caspase-3 inhibitor (1 μM Z-DEVD-FMK, BIOSCIENCES, USA) or medium as a control for 2 h and then treated with BP/LPPC (60 μg/mL for 6 h) or BP (140 μg/mL for 24 h). The level of the Pro-Caspase-3 protein was detected using western blotting.

### 3.10. Synergistic Effects of BP Combined with a Clinical Drug

The analysis of synergistic effects is described below. HT-29 cells (5 × 10^3^ cells) were incubated in 96-well culture plates overnight and treated with BP/LPPC (0, 2.5, 5, 10, 20 and 40 μg/mL) combined with 1 μg/mL 5-FU or 5-FU (0, 0.25, 0.5, 1, 2 and 4 μg/mL) combined with 10 μg/mL BP/LPPC for 24 h. Cell viability was detected using the MTT assay. The combination index (CI) = [(drug A + B) IC_50_/(drug A) IC_50_] + [(drug A + B) IC_50_/(drug B) IC_50_]. The definitions were an additive effect (CI = 1), synergism (CI < 1) and antagonism (CI > 1) [47].

### 3.11. Statistical Analysis

All data are presented as means ± SD (standard deviation). Statistical significance was analyzed using Student’s *t*-test. A *p*-value < 0.05 was considered statistically significant.

## 4. Conclusions

LPPC encapsulation preserved the bioactivity of BP and enhanced the antitumor effect in CRC cells. BP/LPPC with a positive charge triggered cell uptake by activating the clathrin-mediated endocytosis pathway. Moreover, BP/LPPC induced cell cycle arrest at the G_0_/G_1_ phase by altering the levels of the cell cycle regulators CDK4/Cyclin D1. BP/LPPC increased the percentage of cells in the subG_1_ phase, and activated cell apoptosis through the extrinsic (FasL/Fas/Caspase-8) and intrinsic (Bax/Caspase-9) pathways. The combination of BP/LPPC and the clinical drug 5-FU exerted synergistic inhibitory effects, suggesting additional applications in cancer therapy. Furthermore, BP/LPPC has strong potential for development as a clinical drug or adjuvant for CRC therapy.

## Figures and Tables

**Figure 1 molecules-25-02394-f001:**
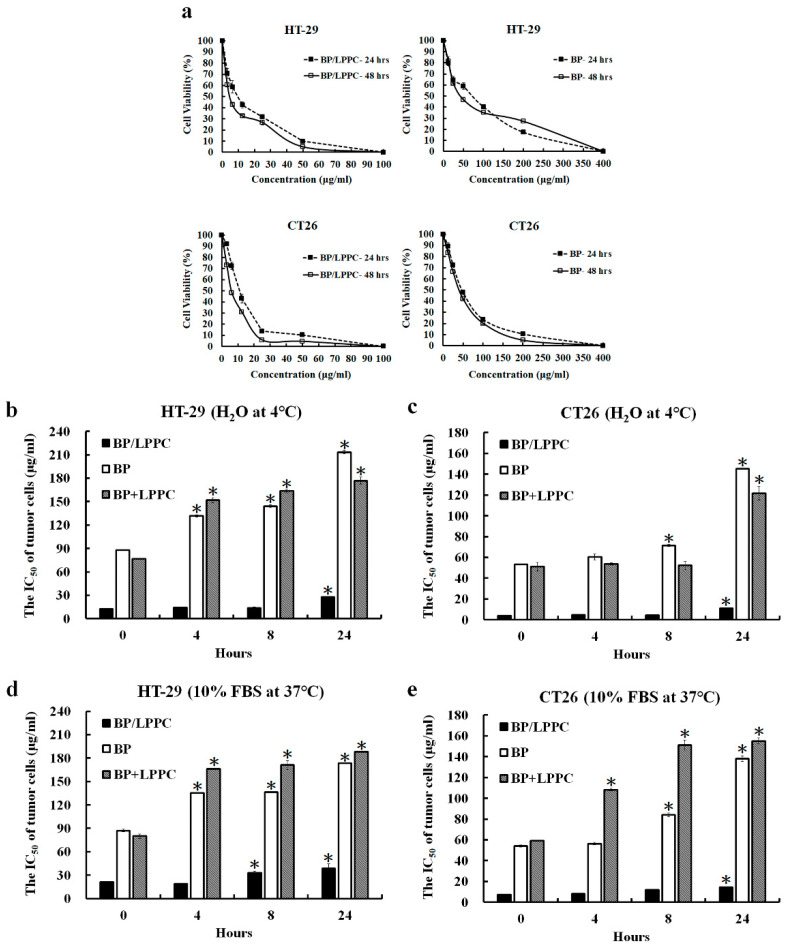
Protective effect of LPPC encapsulation on BP activity. (**a**) Growth inhibition curves of BP/LPPC- or BP-treated cells created based on the results of the MTT assay. (**b**–**e**) BP/LPPC, BP or BP + LPPC were dissolved in H_2_O at 4 °C or 10% FBS in PBS at 37 °C and incubated for 0, 4, 8 and 24 h. HT-29 and CT26 cells were treated with the preincubated BP/LPPC, BP or BP + LPPC and IC_50_ values were calculated using the MTT assay. BP activity was determined by assessing the cytotoxicity toward colorectal cancer cells. * *p* < 0.05 compared with the 0 h in each group.

**Figure 2 molecules-25-02394-f002:**
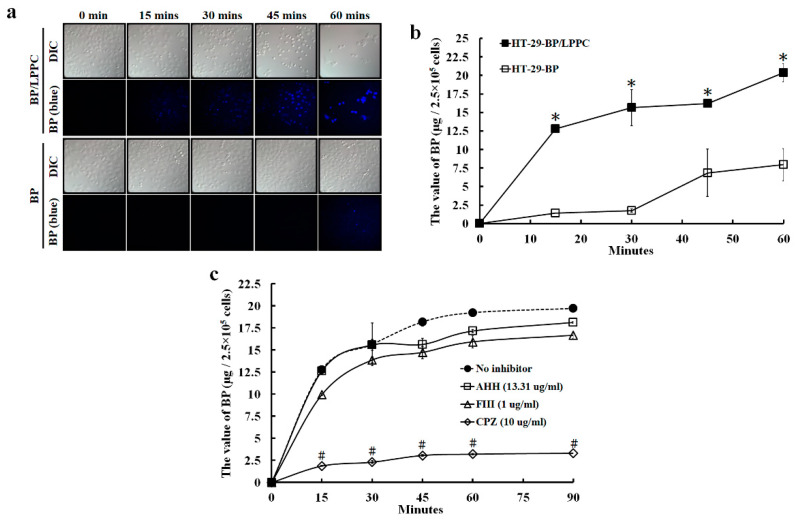
LPPC encapsulation promoted the cellular uptake of BP via the clathrin-mediated endocytosis pathway. (**a**) HT-29 cells were treated with BP/LPPC (50 μg/mL) or BP (50 μg/mL) for 0, 15, 30, 45 or 60 min and the cellular uptake of BP (blue fluorescence) was observed using an upright fluorescence microscope. (**b**) HT-29 cells were incubated with BP/LPPC (50 μg/mL) or BP (50 μg/mL), BP was extracted with phenol-chloroform, and BP levels in cells were determined using a fluorescence spectrophotometer to quantify cellular uptake. * *p* < 0.05 compared with the BP group. (**c**) HT-29 cells were pretreated with the endocytosis inhibitors AHH (13.31 μg/mL), FIII (1 μg/mL) and CPZ (10 μg/mL) for 1 h; then, cells were treated with BP/LPPC (50 μg/mL) and the BP levels in cells were determined as described above. ^#^
*p* < 0.05 compared with the control.

**Figure 3 molecules-25-02394-f003:**
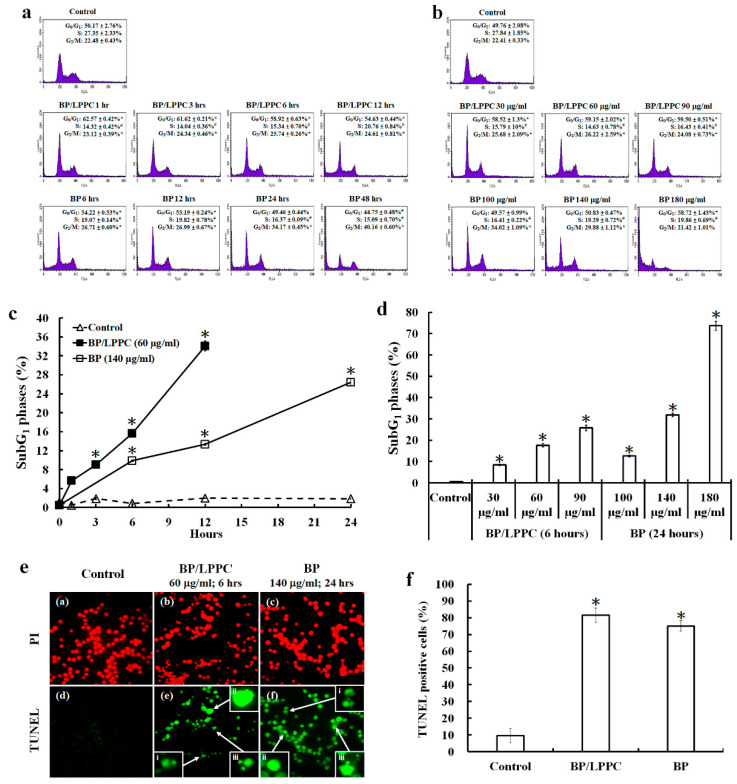
BP/LPPC induced cell cycle arrest and apoptosis in colorectal cancer cells. HT-29 cells were treated with BP/LPPC (30, 60 and 90 μg/mL) for 0, 1, 3, 6 or 12 h or BP (100, 140 and 180 μg/mL) for 0, 6, 12, 24 or 48 h. The treated cells were analyzed for the cell cycle distribution (**a**,**b**), percentage of SubG_1_ (**c**,**d**) and TUNEL staining (**e**). Chromatin condensation (i), DNA fragmentation (ii) and apoptotic bodies (iii) were observed under a fluorescence microscope (400×). (**f**) Statistical analysis of TUNEL assay. The results are presented as the mean ± SD. * *p* < 0.05 compared with the control, with a significant increase. ^#^
*p* < 0.05 compared with the control, with a significant decrease.

**Figure 4 molecules-25-02394-f004:**
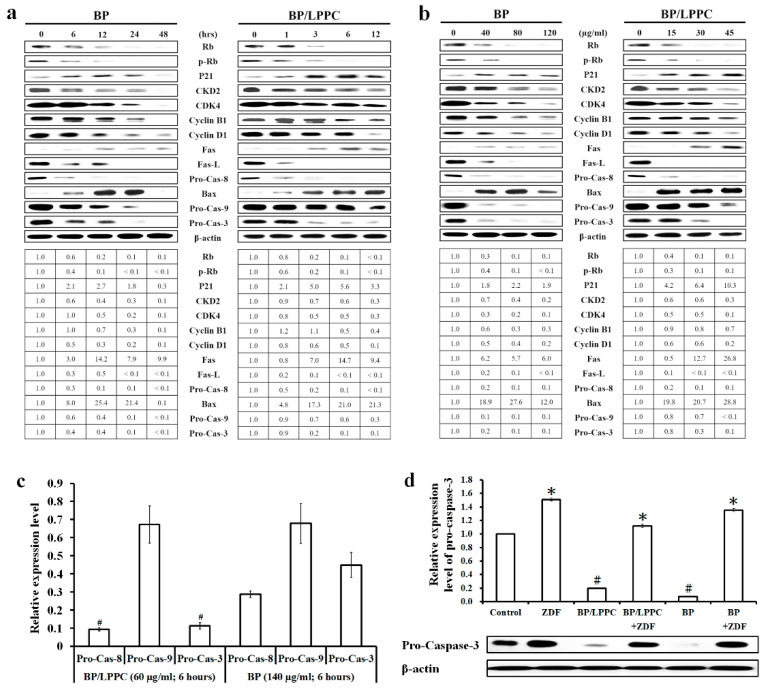
The molecular mechanism underlying the antitumor effects of BP/LPPC in HT-29 cells. (**a**,**b**) HT-29 cells were treated with BP/LPPC or BP at different doses and for different times, and the levels of cell cycle- and apoptosis-related proteins were detected using western blotting. (**c**) Comparison of the levels of the Pro-Caspase-8, 9 and 3 proteins between cells treated with BP/LPPC (60 μg/mL) and BP (140 μg/mL) at 6 h. ^#^
*p* < 0.05 compared with the BP group, with a significant decrease. (**d**) Cells were pretreated with the Caspase-3 inhibitor Z-DEVD-FMK (1 μM), and then treated with 60 μg/mL BP/LPPC (6 h) or 140 μg/mL BP (24 h). The level of the Pro-Caspase-3 protein was analyzed using western blotting. ^#^
*p* < 0.05 compared with the control, with a significant decrease. * *p* < 0.05 compared with no inhibitor pretreatment in the BP/LPPC or BP group, with a significant increase.

**Figure 5 molecules-25-02394-f005:**
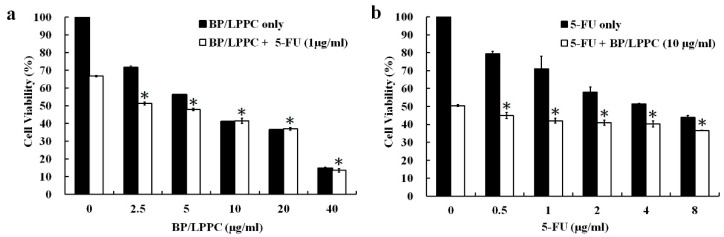
Synergistic effect of BP/LPPC combined with 5-FU. HT-29 cells were treated with (**a**) BP/LPPC (0, 2.5, 5, 10, 20 and 40 μg/mL) combined with 5-FU (1 μg/mL) or (**b**) 5-FU (0, 0.5, 1, 2, 4 and 8 μg/mL) combined with BP/LPPC (10 μg/mL) for 24 h. * *p* < 0.05 compared with the control in the drug combination group.

**Table 1 molecules-25-02394-t001:** IC_50_ values of BP/LPPC against different colorectal cancer cells.

Cell Line	Tumor Type	BP/LPPC	BP	BP/Liposome	5-FU
CRC cells					
HT-29	Human colorectal adenocarcinoma	9.61 ± 2.97 ^a,b^	73.91 ± 2.98	139.33 ± 2.32	>10
CT26	Mouse colorectal adenocarcinoma	11.01 ± 3.96 ^a,b,c^	47.87 ± 2.30	69.61 ± 1.74	1.25 ± 1.86
Normal cells					
SVEC	Mouse vascular endothelial cell	24.15 ± 0.40 ^a,c^	97.48 ± 4.80	ND	7.33 ± 2.39
MDCK	Canine kidney epithelial cell	26.74 ± 3.82 ^a,b^	116.62 ± 0.73	148.69 ± 12.21	>10

Notes: Values are presented as the mean ± SD (μg/mL) at 24 h. ND: not detected. ^a^ Significant difference: BP/LPPC treatment compared with the BP treatment (*p* < 0.05). ^b^ Significant difference: BP/LPPC treatment compared with the BP/Liposome treatment (*p* < 0.05). ^c^ Significant difference: BP/LPPC treatment compared with the 5-FU treatment (*p* < 0.05).

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
