# Peer review of "Antitumor Effects of N-Butylidenephthalide Encapsulated in Lipopolyplexs in Colorectal Cancer Cells"

_molecules, 2020, doi:10.3390/molecules25102394_

Round 1

Reviewer 1 Report

The manuscript demonstrates the effect of liposome encapsulated n-Butylidenephthalide on inhibition of cell growth and induction of apoptosis in human colorectal cancer cells. The authors have assembled polycationic liposomal polyethylenimine and polyethylene glycol complex as a drug carrier that preserve the compound and increases uptake in cancer cells. The authors have compared the effect of drug encapsulated nanoliposome on cancer cell viability, clearance, cell cycle arrest and induction of apoptosis. Furthermore, the authors provided a molecular approach and alteration in the number of proteins regulating cell cycle arrest and apoptosis and their modification in nanoliposome. Lastly, the authors have shown that a combination of liposome encapsulated n-Butylidenephthalide and 5-fluorouracil offer a synergistic effect on colorectal cancer cells at a reduce dose of standard chemotherapeutic drug.

The study is well-intentioned and presented uses methods, which are straightforward, with which the authors have much experience. The novelty of the manuscript is diminished because the group has previously published the concept using n-Butylidenephthalide encapsulated polycationic liposome on B16/F10 melanoma cells. A suggestion to improve the quality of the manuscript is to provide additional discussion of n-Butylidenephthalide and its effectiveness as anticancer agent.

Author Response

Dear Dr.

   We appreciate your expert reviewers for the constructive critiques and comments, and rewrote additional discussion of n-Butylidenephthalide and its effectiveness as anticancer agent, and the content is as following:

     Line 226-242 (high-lighted with yellow ink in full text): CRC is one of the most common malignancies worldwide with particularly high incidences in developed countries [44]. Surgical ablation, radiotherapy and conventional chemotherapy are the major strategies for treatment of CRC, but none of these could be completely effective because of general toxicity and size effects. Therefore, it is urgently developing new therapeutic agents with better efficiency and less side effects in the treatment of CRC. Recent studies demonstrated that liposomes improves efficiency of drugs by increasing the rate of drug uptake, maintaining drug stability and slowly releasing drugs [19-23]. In this study, we demonstrated that LPPC encapsulation protected BP activity, enhanced cellular uptake and increased selection to CRC cells, which improved efficiency of BP treatment. According to reports, cell cycle deregulation is one of the main hallmark features of cancer cells. Loss of cell cycle checkpoints before completing DNA repair will activate the apoptosis cascade, thereby causing cell death [45, 46]. Therefore, drugs which induce apoptosis or cell cycle arrest would be an excellent source of new anti-cancer agents. The results demonstrated that BP/LPPC induced cell cycle arrest at the G0/G1 phase by up-regulation of p53 and p21 proteins expression resulted in altering the levels of the cell cycle regulators CDK4/Cyclin D1. Moreover, BP/LPPC increased the percentage of cells in the subG1 phase, and activated extrinsic (FasL/Fas/Caspase-8) and intrinsic (Bax/Caspase-9) apoptosis pathways resulted in cell death. These findings further confirm the therapeutic potential of BP/LPPC in CRC.

Reviewer 2 Report

The formulation used by Authors is a lipopolyplexe not a nanoliposome.

Also LPPC are above 200 nm and nanoliposomes are below 50 nm.

Authors cannot cite ref 37 as in that paper the drug is adsorbed not encapsulated.

A scheme of LPPC should be provided.

In Fig3, all pahses need to be quantitated not just subG1, with statistical analysis.

Statistical analysis of TUNEL assay should be provided.

LPPC preparation needs to be described.

For encapsulation and free drug experiments, a group of BP + empty LPPC needs to be added.

Authors need to perform biocompatibility assays (platelet agregation and hemolysis), provide stability and release profile of LPPC.

As Authors only apply the same formulation as in refs 38 and 39 to another cancer cell lines, in vivo experiments are required.

Author Response

Dear Dr.

    We appreciate your expert reviewers for the constructive critiques and comments. All the suggestions and all answers of reviewer’s comments were high-lighted with yellow ink. Responses to the specific comments are described as follows:

Q1. The formulation used by Authors is a lipopolyplex not a nanoliposome. Also LPPC are above 200 nm and nanoliposomes are below 50 nm.

We thank for the reviewer’s suggestion and agreed to change the title to “The anti-tumor effects of n-Butylidenephthalide encapsulated in lipopolyplexs in colorectal cancer cells”

Q2. Authors cannot cite ref 37 as in that paper the drug is adsorbed not encapsulated.

Line 78 (high-lighted with yellow ink in full text):

We agree the reviewer’s suggestion and remove the citation ref 37 in this section.

Q3. A scheme of LPPC should be provided.

Line 257-258 (high-lighted with yellow ink in full text):

According the reviewer’s suggestion, we rewrote this part and cite references in materials and methods.

Q4. In Fig3, all phases need to be quantitated not just subG1, with statistical analysis.

Line 157-160 (high-lighted with yellow ink in full text):

We thank the reviewer’s suggestion and the quantitation of G0/G1, S and G2/M phase with statistical analysis were described as following:  

Cells were treated with BP/LPPC (30, 60 or 90 μg/ml) or BP (100, 140 or 180 μg/ml) and the cell cycle was analyzed by monitoring the FL2 intensity using a FACScan instrument to examine the anti-tumor mechanism. Treatment with 60 μg/ml BP/LPPC (1-12 hours) and 140 μg/ml BP (6-12 hours) induced cell cycle arrest at the G0/G1 phase (62.57 ± 0.42% in BP/LPPC at 1 hour; 54.22 ± 0.53% in BP at 6 hours, Figure 3a). In addition, BP/LPPC (30, 60 and 90 μg/ml) and BP (180 μg/ml) induced G0/G1 phase arrest after treatment for 24 hours (Figure 3b).

Q5. Statistical analysis of TUNEL assay should be provided.

Line 165-167 (high-lighted with yellow ink in full text):

We agree the reviewer’s recommendation and present statistical analysis of TUNEL assay in this section and the content is as following:

Treated cells were TUNEL-positive (81.53 ± 4.33% in BP/LPPC; 75.10 ± 3.15% in BP) and exhibited an apoptotic morphology, including chromatin condensation, DNA fragmentation and apoptotic bodies (Figure 3e, f).

Q6. LPPC preparation needs to be described.

Line 257-258 (high-lighted with yellow ink in full text):

According the reviewer’s recommendation, we rewrote this part and cite references in materials and methods.

Q7. For encapsulation and free drug experiments, a group of BP + empty LPPC needs to be added.

Line 102-112 (high-lighted with yellow ink in full text):

We agree the reviewer’s suggestion and add BP + empty group in the result (Fig1), and the content is as following:

The drug designs included LPPC encapsulation (BP/LPPC group), no LPPC encapsulation (BP group) and BP + empty LPPC (BP+LPPC group) to determine whether the activity of BP was protected after encapsulation in LPPC. In HT-29 and CT26 cells, BP/LPPC showed greater cytotoxicity (IC50 = 11.06 ± 0.37 - 27.60 ± 1.10 μg/ml, 24 hours) than BP (IC50 = 145.32 ± 0.35 - 213.41 ± 2.04 μg/ml, 24 hours) and BP+LPPC (IC50 = 121.6 ± 6.64 - 176.81 ± 4.56 μg/ml, 24 hours) after storage at 4°C in H2O (Figure 1b, c). In addition, BP/LPPC (IC50 = 14.57 ± 0.15 - 38.38 ± 5.91 μg/ml, 24 hours) also displayed greater cytotoxicity than the BP group (IC50 = 138.03 ± 2.88 - 173.25 ± 0.52 μg/ml, 24 hours) and BP+LPPC (IC50 = 155.02 ± 2.96 - 188.14 ± 0.3 μg/ml, 24 hours) after storage at 37°C in PBS containing 10% FBS (Figure 1d, e). The IC50 value was rapidly increased in the BP group and BP+LPPC group after an incubation at 4°C or 37°C for 4-24 hours but was not obviously altered in the BP/LPPC groups.

Q8. Authors need to perform biocompatibility assays (platelet aggregation and hemolysis), provide stability and release profile of LPPC.

We thank the reviewer’s suggestion. In our previous study showed that BP/LPPC was analyzed stability and release profile in different environment such as different pH, oxidation, protein rich in vitro, and biodistribution of BP in tumor and organs in vivo. The biodistribution of BP showed LPPC delivers most of the BP to the tumor area. Unencapsulated BP also accumulates in the tumor, but they still partially accumulate in the liver (ref 37, 39). In our another study undergoing submission, F344 rats were treated with DMSO, BP/LPPC or BP by i.v. injection with one dose and collected blood cells analyzed by using a hematology analyzer at 0, 6 and 12 hours . The results revealed that it was no significant difference between groups (data in following table), indicated there is no obvious platelet aggregation or hemolysis after BP/LPPC treatment.

Platelet (×103/μL)

Red Blood Cells (×106/μL)

Hours

0 h 6 h 12 h 0 h 6 h 12 h

Vehicle

511.67 ± 17.05 437 ± 14.73 496 ± 6.25 7.89 ± 0.34 7.44 ± 0.24 7.06 ± 0.34

BP/LPPC

470.67 ± 26.36 516 ± 10.02 412.67 ± 56.71 8.29 ± 0.08 7.82 ± 0.3 7.03 ± 0.19

BP

497.33 ± 12.41 465.67 ± 4.48 453.33 ± 21.79 8.46 ± 0.31 7.03 ± 0.26 7.14 ± 0.18

Q9. As Authors only apply the same formulation as in refs 38 and 39 to another cancer cell lines, in vivo experiments are required.

We thank the reviewer’s recommendation. In our previous study showed that BP/LPPC suppressed GBM growth in xenograft and orthotopic animal model (ref 37, 39). In xenograft animal model, mice bearing DBTRG-05MG tumors were treated with BP (100 mg/kg), empty LPPC, or BP/LPPC (containing 100 mg/kg BP) once every 2 days by IV injection. After 14 days, the animals treated with BP/LPPC showed a significant suppression of DBTRG-05MG tumor growth (inhibition of tumor growth by ~85%) compared with the untreated controls, the nonencapsulated BP-treated animals, and the empty LPPC-treated animals. In orthotopic animal model, MRI data revealed that the in-situ tumor volumes in the BP/LPPC-treated group were smaller than those in the control group and in 60% mice tumors had completely disappeared. In this study, we demonstrated that BP/LPPC inhibited growth of CRC cells and cause cell death. Therefore, the same formulation of BP/LPPC was implied similar anti-cancer effects in CRC in vivo. In our future work, we will design the antimetastatic experiment of BP/LPPC in CRC in vitro and in vivo.

Round 2

Reviewer 2 Report

Authors answered all cmments so I recommend publication of the revised paper.